# A framework for testing independence between lane change and cooperative intelligent transportation system

Mohammed Elhenawy[☯], Sébastien Glaser[☯], Andy Bond[☯], Andry Rakotonirainy[☯], Sébastien Demmel[☯], Mahmoud Masoud[ID]*[☯]

Centre for Accident Research & Road Safety, Queensland University of Technology, Brisbane, QLD, Australia

☯ These authors contributed equally to this work.
* Mahmoud.masoud@qut.edu.au

**Data Availability Statement:** The pre-processed data in excel sheet is provided and attached with the documents. Also, the raw (the UAH-DriveSet) data is available publically at: http://www.robesafe.

## Abstract

Cooperative Intelligent Transportation Systems (C-ITS) are being deployed in several cities around the world. We are preparing for the largest Field Operational Test (FOT) in Australia to evaluate C-ITS safety benefits. Two of the safety benefit hypotheses we formulated assume a dependency between lane changes and C-ITS warnings displayed on the Human Machine Interface (HMI) during safety events. Lane change detection is done by processing many predictors from several sensors at the time of the safety event. However, in our planned FOT, the participating vehicles are only equipped with the vehicle C-ITS and the IMU. Therefore, in this paper, we propose a framework to test lane change and C-ITS dependency. In this framework, we train a random forest classifier using data collected from the IMU to detect lane changes. Consequently, the random forest output probabilities of the testing data in case of C-ITS and control are used to construct a 2x2 contingency table. Then we develop a permutation test to calculate the null hypothesis needed to test the independence of the lane change during safety events and the C-ITS.

## Introduction

The United States was the first country to introduce the concept of intelligence into transportation systems in the 20th century [1]. Nowadays, the research and development of ITS have moved to other countries such as Japan, the European Union, South Korea, Australia, and Qatar. In the 1970s, the United States started the Electronic Route Guidance System (EGRS), which is considered one of the initial ITS applications. Later, in 1991, the Integrated Surface Transportation Efficiency Act (ISTEA) was enacted. The Transportation Equity Act for the 21st Century TEA-21 followed ISTEA.

C-ITS is the technology that enables the components of ITS (road users, vehicles, roadside units) to communicate and cooperate via wireless networks. The interaction between ITS components is based on several standards developed by different organizations such as European Committee for Standardization (CEN) [2], the International Organization for Standardization

uah.es/personal/eduardo.romera/uah-driveset/#download

**Funding:** The authors would like to acknowledge the support of the iMOVE Cooperative Research Centre (CRC), where this work is funded under grant number 1-002.

**Competing interests:** The authors have declared that no competing interests exist.

(ISO) [3], the European Telecommunications Standards Institute (ETST) [4], depending on the jurisdiction. An important step towards C-ITS was the cooperation between governmental transportation departments and vehicle manufacturers around the world to enable vehicle-to-vehicle (V2V) and vehicle-to-infrastructure (V2I) communications. This enabled C-ITS to build an integrated system of people, roads, and vehicles by applying and integrating communications, computers, and other technologies in the field of transportation. Integrating information and technologies can establish a large, full-functioning, real-time, accurate, efficient, and safe transportation system.

There are many C-ITS projects around the world [5–8] including:

- Cooperative Intelligent Transport Initiative (CITI) in Australia,

- Cooperative Intelligent Transport Systems (C-ITS) Pilot which is the main component of the Cooperative and Automated Vehicle Initiative (CAVI) in Australia[9],

- Cooperative Mobility Pilot on Safety and Sustainability Services for Deployment (COM-PASS4D)—a C-ITS deployment in Europe,

- ETC2.0 Project in Japan, and,

- Safe and Intelligent Mobility—Test Field (SIM-TD) in Germany.

Recent research uses lane changes as a safety indicator to evaluate the safety benefits of C-ITS [10]. The challenge in such studies is the limited number of sensing modalities so that the trained machine learning models yield relatively low sensitivity and accuracy. The following paragraphs will show why lane change detection/prediction is an important surrogate measure of safety. We will briefly describe different approaches used for lane change detection/prediction and list the used sensing modalities.

Scene understanding, scene interpretation, and the prediction of the associated driver behaviour are critical tasks for advanced driver assistance systems, semi-autonomous, and fully autonomous driving vehicles. Lane change detection/prediction is one pattern of behaviour which has attracted the attention of researchers because of two reasons. First, it is important for semi-autonomous and fully autonomous driving. For example, lane change prediction of other non-autonomous vehicles is important for risk assessment for safe driving. Second, safely changing lane in traffic requires the driver to look forward and sideways to adjacent lanes. Looking in these two directions makes a lane change a complicated manoeuvre which could contribute to crashes. Fitch and Hankey analysed the 2009 National Automotive Sampling System General Estimates System and found that 7.44 per cent of all car crashes are due to lane changes [11]. The different approaches used for lane change detection/prediction are based on different sensing modalities, including mono and stereo vision, LIght Detection and Ranging (LIDAR), car odometry, inertial measurement unit (IMU), global positioning system (GPS), electrocardiogram (ECG), galvanic skin response (GSR), and respiration rate (RR).

Woo et al. [12] installed sensor system in the subject vehicle to predict the target vehicle trajectory and hence detect the lane change of the target vehicle; the system consisted of a position sensor (RT3003) and six laser scanners (Ibeo LUX). This system enabled the subject vehicle to acquire its position, the position of other vehicles, and their velocity. The authors pre-processed this information to construct the input features vector to the support vector machine (SVM) driving-intention estimation model. The SVM output is used to predict the trajectory of the target using the potential field method. Consequently, a lane change can be detected with high accuracy, on average, 1.74 s before the target vehicle crosses the centre line. Mandalia and Salvucci [13] applied SVM to data collected from an instrumented vehicle to detect lane change intentions. They tried several subsets of features that include lane position,

heading, steering angle, and acceleration. Their proposed approach achieved the best recognition result with high accuracy when using lane position features at different longitudinal distances from the driver's vehicle. Salvucci [14] proposed mind-tracking architecture for detecting driver lane changes. The proposed architecture simulates a set of possible driver intentions to predict their resulting steering-wheel angle and accelerator depression behaviours. Then the observed driver behaviour is matched with the predicted behaviour using a Gaussian kernel to find the most probable driver intention. The proposed architecture requires measuring two visual features in addition to the minimum time headway to either the lead vehicle (in case of lane-keeping) or to the lead vehicle in the destination lane (in case of lane changing). McCall and Trivedi [15] developed a novel "video-based lane estimation and tracking" (VioLET) system. The system is based on steerable filters to detect lane-markings. The proposed system is efficient in detecting circular-reflector markings, solid-line markings, and segmented-line markings in different lighting and road conditions. It is suitable for lane detection in different traffic situations including overpasses and tunnels. Xuan and Coifman [16] used the Differential Global Positioning System (DGPS) data to detect lane change without high-resolution maps. They started by creating a reference trajectory by estimating the median of multiple probe vehicle trajectories through a study corridor. After controlling for mandatory lane change manoeuvres (MLC), the authors created virtual lanes by setting lateral thresholds around the reference trajectory, which were then used for lane change detection. Henning *et al.* [17] compared selected behavioural and environmental indicators that can be used to predict the intention to change lane. The authors extracted these indicators from drivers, vehicles, and environment data collected in a field study. They reported that the glance to the left outside mirror is a potential indicator to predict the driver intention to change lanes.

Jeong *et al.* [18] used state of the art deep convolutional neural network to classify the adjacent lanes from two rear cameras. The proposed approach achieved 96.98% classification accuracy which makes it a promising assistance system for lane-change decisions for human drivers and autonomous vehicles. Jang *et al.* [19] proposed using eye movements and the pupil size of the driver to predict lane change intention. Extracted features from eye moments data are used to train a discriminative classifier to identify the probable lane change. Zheng and Hansen [20] explored the effectiveness of using steering angle and vehicle speed extracted from the CAN-bus for lane change detection. They proposed a machine learning-based segmentation and classification algorithm to detect lane-change and achieved 80.36% classification accuracy for lane-change-left and 83.22% for lane-change-right. Schlechtriemen *et al.* [21] used data collected by a vehicle equipped with a front-facing stereo camera and several radar sensors to obtain a 360 field of view. The data was used to extract several features, which were modelled using a mixture of Gaussian distributions. Then a naïve Bayesian classifier was used to estimate the probabilities of the left lane change, right lane change, and lane-keeping for new incoming feature vectors. Kasper *et al.* [22] used an object-oriented Bayesian network for the recognition of lane changes. The authors investigated the usage of lane-related coordinate systems and occupancy schedule grids for all modelled vehicles. Their proposed approach recognised a lane change 0.6s earlier than a standard adaptive cruise control (ACC) system.

There are two papers proposing using physiological signals from the driver to predict their intention to change lane. Those signals are the electrocardiogram (ECG), galvanic skin response (GSR), and respiration rate (RR). In the first paper, Murphey *et al.* [23] computed a set of biologically meaningful signals using the ECG, GSR and RR signals. Then a subset of the computed biological signals was selected using the Granger causality test. This subset of signals was used as input to a neural network for classification of lane changing and lane-keeping [23]. In the second paper, Gao *et al.* [24] used these three signals as an input to a proposed novel Group-wise Convolutional Neural Network to predict lane change intention.

Experimental results using a data set collected from five different drivers showed that the proposed approach had a promising prediction accuracy.

The present paper is different from earlier research in that it developed a framework consisting of the following:

1. Training a lane change/lane-keeping machine learning classifier using a limited number of sensing modalities (i.e. longitudinal acceleration, lateral acceleration, roll, and yaw)

2. Overcoming the uncertainty of machine learning classifier if it classifies one data point. This occurs by summing up the output probabilities of the trained classifier of the whole collected data (i.e. all data points) to construct the 2x2 contingency table.

3. Estimating the null hypothesis of the test statistic using the proposed permutation test.

This framework enables us to test the independence between the lane change and the C-ITS warning without collecting expensive data modalities such as videos to train accurate machine learning classifier. In other words, the proposed framework allows us to test the hypotheses that assumes C-ITS warnings at safety-critical situation help the drivers changing lane safely compared to the drivers who do not receive C-ITS warnings.

## Problem statement and proposed approach

The authors of this paper are members of a larger team responsible for designing the largest on-road trial in Australia of cooperative vehicles and infrastructure. This FOT will be deployed on the public road from 2020 for a year. One goal of this FOT is evaluating the safety benefits of the Cooperative Intelligent Transport Systems (C-ITS). The FOT will test seven use-cases applications that hopefully will make roads safer and hence contribute towards the Vision Zero goal of no road deaths and serious injuries on Australian roads.

This FOT will include a large number of passengers car which are equipped with C-ITS and IMU devices. Moreover, many C-ITS roadside devices will be deployed on arterial roads and motorways. The installed devices will allow real-time vehicle-to-vehicle and vehicle-to-infrastructure communication to exchange information related to the road and vehicles; this combined information will generate relevant safety warning messages for drivers. In this paper, we are interested in two use-case applications, namely the Slow/Stopped Vehicle (SSV) and The Road Works Warning (RWW).

The Slow/Stopped Vehicle (SSV) event provides warning to the Subject Vehicle (SV) (upstream) which is triggered by a slow or a stopped C-ITS enabled Target Vehicle (TV) (downstream). Every cooperative vehicle is consistently broadcasting cooperative awareness message (CAM) [25] to other cooperative vehicles within communication range. These CAM messages include position, heading, speed, acceleration, and other parameters from the GPS chipset of the emitting vehicle. It is the responsibility of the SV to detect the SSV vehicle hazard by processing received CAM messages. The in-vehicle HMI will then display an SSV warning to the driver, as shown in Fig 1.

The Road Works Warning (RWW) provides an advanced warning to the SV approaching a roadworks zone. The RWW DENM (decentralized environmental notification message) is created and broadcasted by C-ITS-F which interfaces to a Transport and Main Roads road works database service. The DENM contains the work zone information such as the speed limit, lane closures, and location. The SV will receive the RWW DENM and decide on the relevance of its location to the work zone and, in particular, the approach zone to the road works. When the location of the road-work becomes relevant, the HMI will display an RWW warning in addition to the speed limit and lane closure (if any) to the driver, directing them to take suitable action, as shown in Fig 2.

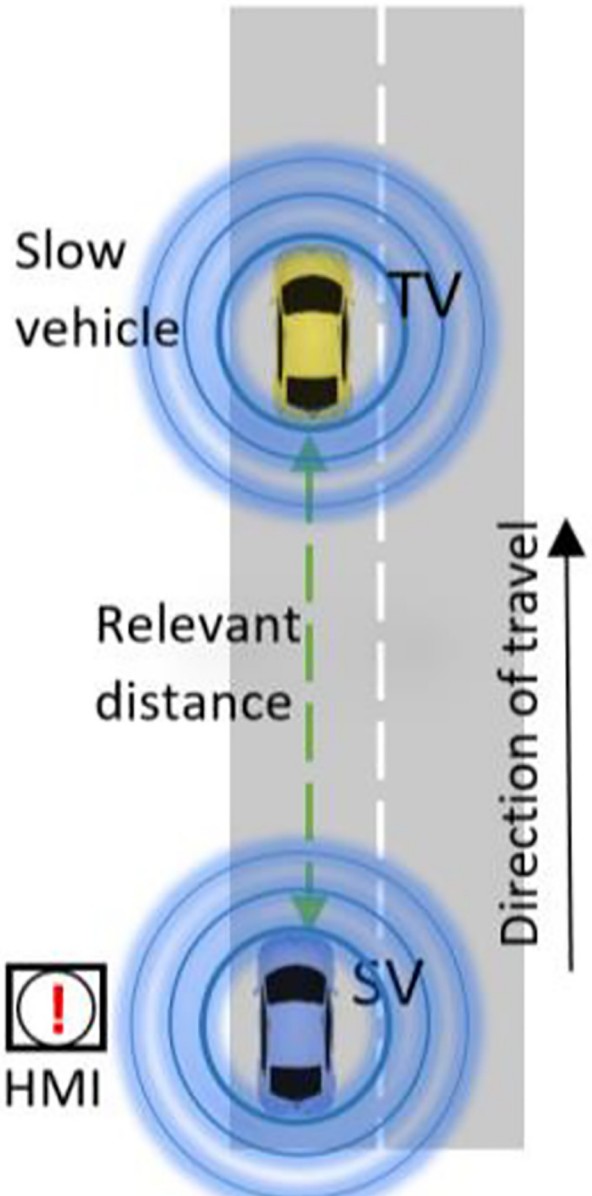

**Fig 1. General operation of SSV.**

In the non-C-ITS environment, SSV is similar to lane closure due to roadworks because both can cause safety risk and traffic capacity drop. In the case of lane closure due to roadworks, the driver will see the posted warning before the position of the lane closure but this kind of warning does not exist in the SSV case. In the above two cases, in the previous two figures, the event speed is zero; however, the speed of SSV event is not zero. Thus, to our mind, it has less safety risk compared to the zero speed safety events.

In the case of Slow/Stopped Vehicle and lane closure from roadworks, we test the hypotheses that the driver will change lane early before the location of the lane closure. In other words, we need to test the independence of the lane change and the SSV warning as well as the independence of the lane change and the lane closure warning for RWW.

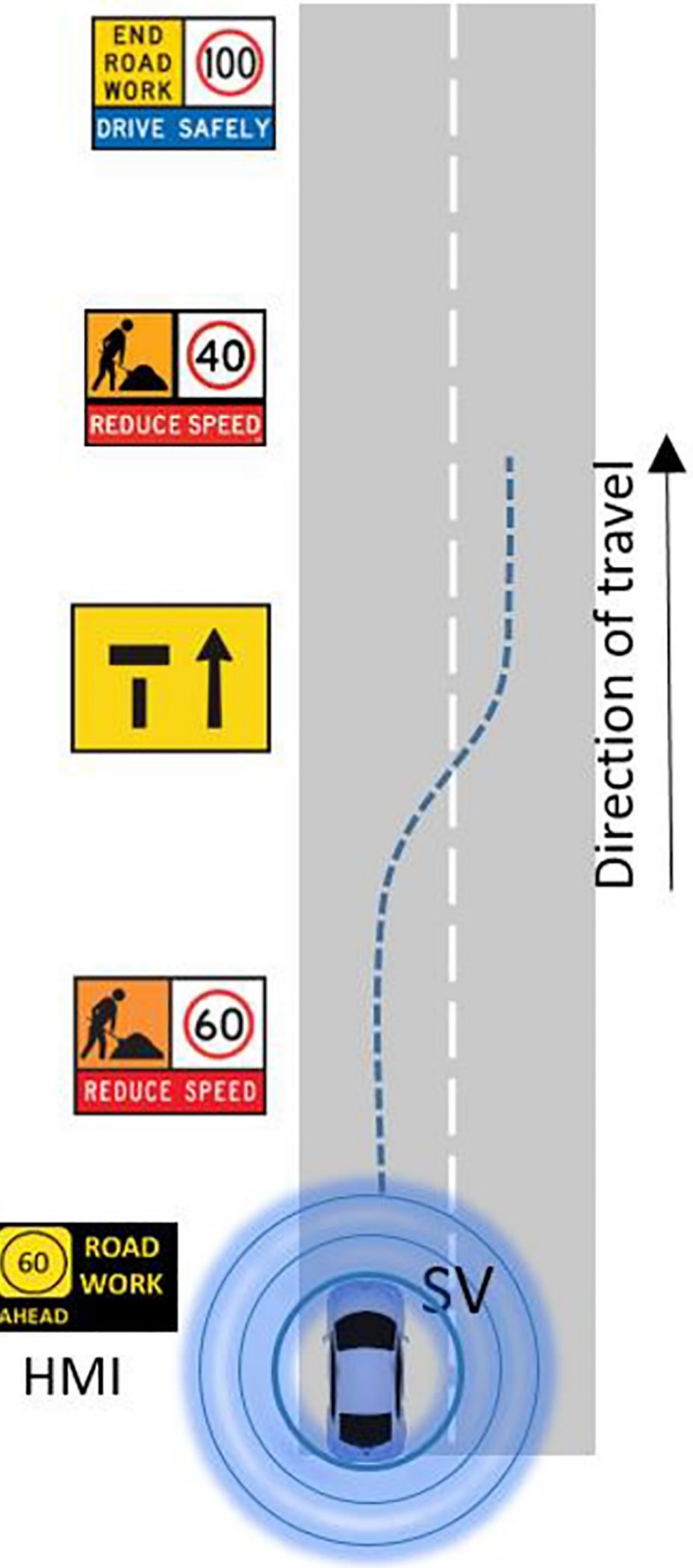

**Fig 2. General operation of RWW.**

In the remaining of this paper, C-ITS warnings or application will only mean SSV or RWW. Moreover, lane change and early lane change will be used interchangeably. Testing the independence of the lane change and C-ITS warning displayed on the HMI will require constructing a 2x2 contingency table using the IMU data collected from the C-ITS vehicles and non-C-ITS (control) vehicles during the safety events related to a particular C-ITS application. Then using the chi-square statistical test to check the independence of lane changing and C-ITS warning. This approach looks straightforward; however, estimating the elements of the 2x2 contingency table is not trivial because we only have IMU data.

In this paper, given a set of labelled (lane change/lane-keeping) IMU reading, we proposed using the Fast Fourier Transform to estimate power spectrum for each of the IMU signals (Acceleration in Y [Gs], Acceleration in Z [Gs], Roll [degrees], and Yaw [degrees]) within a window of a predefined length. In a real scenario, this window will be started at the onset time of the C-ITS warning. Consequently, we concatenate the power spectrum of all signal of each example to form the corresponding features vector, which is a row in the training matrix. The training data is used to build a Random Forest model to classify the new unseen and unlabelled features vectors. For each unlabelled features vector, the classification output could be one and zero (which we called a "hard" decision), or the probability of each label (which we called a "soft" decision). Finally, we use the output of the classifier to construct the 2x2 contingency matrix and test the C-ITS and lane change and the direction of the association. In the following sections, we will describe in details each of the above steps and show the experimental results.

## UAH-DriveSet dataset

In this section, we describe the dataset used to train and test the proposed framework to evaluate the safety benefits in the planned FOT. At the time of writing, we are in the planning phase for the FOT, when we proposed the hypotheses based on the FOT research questions. Moreover, the sensors measures and the models/frameworks to test the hypotheses are identified. Because we proposed a new framework, testing is needed before starting the data collection from the FOT; for that, we used the UAH-DriveSet [26]. The UAH-DriveSet it exhibits the same characteristics as the dataset we will collect for the FOT. This dataset was collected using six different drivers and vehicles. The dataset has more than 500 minutes of naturalistic driving on highways and arterial roads. The data set is collected with the smartphone application DriveSafe, which uses the smartphone's inertial sensors, GPS, camera, and internet access. The smartphone with the DriveSafe is put up on the centre of the windshield and its rear camera is facing the road. DriveSafe executes its calibration routine at startup in order to make the smartphone perpendicular to the ground such that the Y-axis aligns with the lateral axis of the vehicle and Z-axis with the longitudinal axis of the vehicle. The experimenters used another phone and put it up on the windshield beside the first phone to capture the video of the road during the drive session. For each trip, the available data includes lane changes list, raw accelerometer data at 10 Hz, and the video of the trip, which is reduced to generate the lane changes list. The lane changes list is used as the ground truth labels in our experiment.

In the context of the C-ITS safety benefits evaluation, we are interested only in detecting lane changes versus lane keeping. Thus we consider slow right, fast right, slow left, and fast left lane changes one single class (lane change) and assign them the same label.

## Methods

In this section, we will discuss the details of the features extraction and the method used to test the independence between lane changes and the C-ITS, as well as the method used to test the direction of the association between the lane changes and the C-ITS.

   

## Random forest

The random forest (RF) [27] is an ensemble approach that is very efficient in dealing with continuous response and labels. The most significant advantage of the RF approach is that it does not suffer from overfitting because of the Law of Large Numbers, where the error rate of RF will get smaller as more CARTs are added. By adding more CATRs, the error rate will be saturated but will never get larger no matter how many CARTs are added.

The main concept behind RF is building a large group of weak models that will give a resultant strong model. The RF is a large group of un-pruned decision trees with a randomised selection of predictors' subset at each split. Each un-pruned decision tree is trained using a bootstrap sample from the original dataset. The well-known machine learning technique called Classification And Regression Tree (CART) [28, 29] is one of the common decision trees used in RF. An RF starts with the CART which, in ensemble terms, corresponds to the weak model. CART is a greedy and recursive top-down binary partitioning that divides the feature space into sets of disjoint regions. These regions should be pure with respect to the response variable. Given trained RF classifier, the label assigned to a test example is obtained based on the majority votes from all trees, [30, 31].

## Features extraction

We have four channels/sensors. Each one provides us with a time series, namely:

1. Acceleration in Y filtered by KF (Gs)

2. Acceleration in Z filtered by KF (Gs)

3. Roll (degrees)

4. Yaw (degrees)

In order to build a data set that has the lane changing examples, for each time series, we take a short window (2w+1) around the event as shown in Fig 3 and extract features from it. The output of the four windows from the time series is concatenated to form an example of a lane change in the dataset. We created several examples from the same lane change event by shifting the window left and right such that the location of the event inside the window would be different. Creating several examples from the same event has two advantages. First, this data

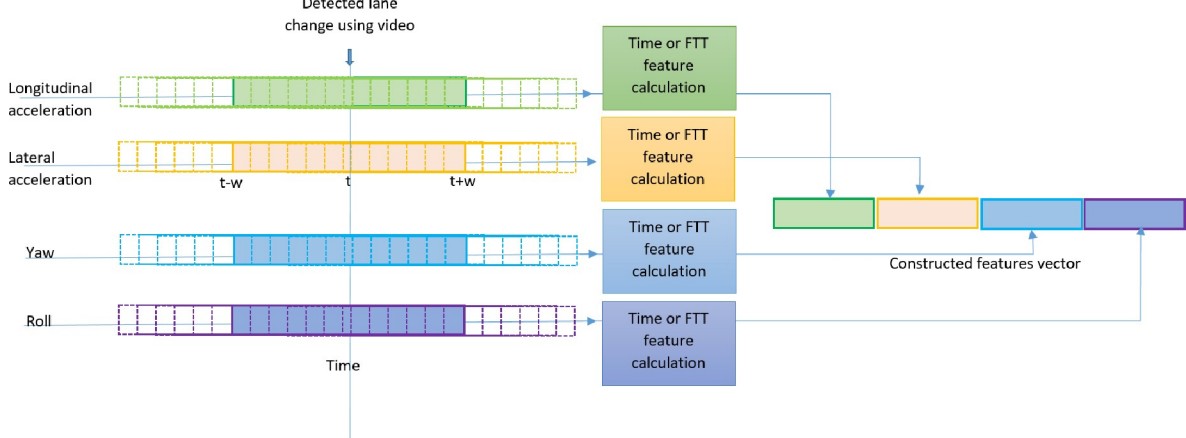

**Fig 3. Illustration of constructing a lane change training example using the time of the event and the raw accelerometer data.**

set will have the event at different locations within the window and hence the model built using this data will be independent of the exact location of the event within the window. Second, we will overcome the problem of a relatively small number of events in the UAH-Drive-Set. We tried two sets of features. The first set is based on calculating the following quantities of each window; note that we applied the measure to both the values inside the window and the difference of the values inside the window (Table 1).

The second set is based on calculating the FFT and then calculating the single side spectrum as the input feature for each time series within the window, as shown in Fig 3.

## Chi-square test

As will be shown in the experimental work section, the sensitivity of the RF classifier is not good and our goal is only to detect if the number of lane changes and the usage of the C-ITS are independent. So instead of using zero/one output of the RF classifier (i.e. hard decision), we use the probabilities of lane change and lane-keeping (i.e. soft decision). The lane change and lane-keeping probabilities of the C-ITS and Non-C-ITS examples are used to construct the 2x2 contingency table. In this case, we are violating the assumption of the cells of the table being drawn from one multinomial distribution or two binomial distributions. However, the cell probabilities can be estimated using the sample mean. In the context of our problem, the explanatory variable is the presence (i.e. C-ITS) or absence (i.e. non C-ITS) of the C-ITS and the response is the lane change or lane-keeping. To test the independence of the number of the lane change and the C-ITS we used the Pearson chi-squared statistic

$$\chi^2 = \frac{\sum_{i,j}(n_{ij} - \mu_{ij})^2}{\mu_{ij}}$$

Where
$n_{ij}$ is the observed cell $(i,j)$
$\mu_{ij}$ is the expected frequency of cell $(i,j)$

Given $n_{1+}$ of C-ITS examples and $n_{2+}$ of non C-ITS examples. We do not know how many lane changes $n_{11}$ or $n_{21}$ and we want to test the independence of the lane change and the C-ITS.

Recall that the given examples are in the form of raw measurement from the IMU. Moreover, we are given a trained RF classifier which has low sensitivity. In this paper, the raw data is processed as described in the above section to extract the input features to the RF classifier. The output of the RF are the probabilities of the lane change and the lane-keeping given certain data example. For the $n_{1+}$ C-ITS examples, the output of the RF is $n_{1+}$ pairs of probabilities. Each pair consists of the probability of lane change and the probability of lane-keeping for certain example in $n_{1+}$. The summation of the $n_{1+}$ lane change probabilities are the observed $n_{11}$ where the summation of the $n_{1+}$ lane keeping is the observed $n_{12}$. Similarly for $n_{2+}$ we have $n_{21}$ and $n_{22}$ which are the lane change and lane-keeping observed numbers in the case of the non C-ITS.

**Table 1. Measures applied to create features.**

| Measure | Measure | Measure | Measure |
|---------|---------|---------|---------|
| Mean(x) | Std(x) | Mean(diff(x)) | Std (diff(x)) |
| Max(x) | Range (x) | Max(diff(x)) | Range (diff(x)) |
| Min(x) | iqr(x) | Min(diff(x)) | iqr(diff(x)) |
| Var (x) | Energy(x) | Var (diff(x)) | Energy(diff(x)) |

The above contingency table can be used to estimate three different probabilities: joint, marginal, and conditional. Let $\pi_{ij} = P(presence\ of\ C-ITS = i,\ early\ lane\ change = j)$ denotes the probability of observing cell $(i,j)$. These set of probabilities form the joint probability of the explanatory variable and the response and they satisfy $\sum_{i,j}\pi_{ij} = 1$. We can get the marginal distribution of the lane change by calculating the column total $\pi_{+1} = \pi_{11}+\pi_{21}$ and $\pi_{+2} = \pi_{12}+\pi_{22}$. Where $\pi_{+1}$ and $\pi_{+2}$ are the marginal probability of lane change and lane-keeping respectively. Since we observed a sample from the population, so we need to estimate the $\pi_{ij}$ using the sample mean $\hat{\pi}_{ij} = \frac{n_{ij}}{N}$, where $N = n_{+1}+n_{+2} = n_{1+}+ n_{2+}$

Moreover, we can use Table 2 to estimate a separate probability distribution for lane change at the given level of the explanatory variable (i.e. C-ITS or Non-C-ITS). These probability distributions are the conditional distribution. The lane change and C-ITS are said to be statistically independent if the population conditional distributions of lane changes are identical at each level of C-ITS. In other words, When these two variables are independent, the probability of lane change is the same at C-ITS and Non-C-ITS.

## The odds ratio test

The odds of lane change are defined as $odds = \frac{\pi}{1-\pi}$ where $\pi$ is the probability of lane change. In the context of our 2x2 table, within the first row (C-ITS) the odds of lane change are $odds_{C-ITS} = \frac{\pi_{C-ITS}}{1-\pi_{C-ITS}}$ and within the second row (non C-ITS) the odds of lane change are $odds_{Non-C-ITS} = \frac{\pi_{Non-C-ITS}}{1-\pi_{Non-C-ITS}}$. Then the odds ratio from the two rows is defined as, $\theta = \frac{odds_{C-ITS}}{odds_{Non-C-ITS}}$. The odds ratio can equal any positive number. The odds ratio $\theta$ equals one when the C-ITS and the lane change are independent ($\pi_{Non-C-ITS} = \pi_{C-ITS}$). So that $\theta = 1$ is the baseline for comparison. When $\theta > 1$ this means that the probability of lane change ($\pi_{C-ITS}$) in the presence of C-ITS is higher than the probability of lane change in the absence of C-ITS ($\pi_{Non-C-ITS}$). When $\theta < 1$ this means that the probability of lane change ($\pi_{C-ITS}$) in the presence of C-ITS is smaller than the probability of lane change in the absence of C-ITS ($\pi_{Non-C-ITS}$).

As shown, the odds ratio has the advantage of showing the direction of the association between the C-ITS and the lane change which is important for evaluating the safety benefits of the C-ITS. In this paper, the sample odds ratio equals the ratio of the sample odds in the two rows $\hat{\theta} = \frac{\frac{\hat{\pi}_{C-ITS}}{1-\hat{\pi}_{C-ITS}}}{\frac{\hat{\pi}_{Non-C-ITS}}{1-\hat{\pi}_{Non-C-ITS}}} = \frac{\frac{n_{11}}{n_{12}}}{\frac{n_{12}}{n_{22}}} = \frac{n_{11}n_{22}}{n_{12}n_{21}}$. As mentioned in the above subsection, the observed cell counts are obtained by summing the output probabilities of the C-ITS and the non C-ITS examples using the RF. Then using the permutation test, we construct the null hypothesis. Consequently, we calculate the p-value of the observed sample odds ratio and drive the conclusion about the direction of the association between the C-ITS and the lane change.

## Permutation test

The basic idea of the permutation test is randomly rearranging the observed sample to generate the sampling distribution of the test statistic under the null hypothesis. In the context of this paper, the null hypothesis is the lane change and C-ITS are independent. Under this hypothesis, all possible permutations are equally likely, and the Pearson chi-squared statistic is computed for one thousand of permutation to estimate its sampling distribution under the independence of lane change and C-ITS. Below are the steps to estimate the p-value using the permutation test:

1. Given the RF outputs of C-ITS and Non-C-ITS examples (original data), construct the 2x2 contingency table;

**Table 2. Illustration of the 2x2 contingency table.**

| | | Response | | |
| --- | --- | --- | --- | --- |
| | | Lane change | Lane-keeping | Total |
| Explanatory variable | C-ITS | $n_{11}$ | $n_{12}$ | $n_{1+}$ |
| | Non-C-ITS (control) | $n_{21}$ | $n_{22}$ | $n_{2+}$ |
| | | $n_{+1}$ | $n_{+2}$ | N |

2. Compute the original test statistics using the original 2x2 contingency table;

3. Randomly permute RF outputs, where we randomly move RF outputs among the C-ITS and Non-C-ITS as shown in Fig 4;

4. Construct the 2x2 contingency table using the permuted data—notice that columns and rows sums are not changed (i.e. marginals are the same as the original data);

5. Compute the test statistics using the 2x2 contingency table of the permuted data;

6. Repeat steps 3, 4, and 5 large number of times to get the distribution of the test statistics under the null hypothesis.

Finally, we compare the original test statistic to the estimated null distribution to estimate the p-value as will be explained in the experimental work section.

## Experimental work

This section is divided into two subsections. In the first subsection, we describe the RF classifier training. In the second subsection, we use the classification output of all examples in the dataset to test the proposed test at different sample and effect size.

## Classification

The time series output of the four selected sensors is used to extract the features needed to train the RF. In this paper, we investigated two sets of features. The first features set is based on the calculation of the summary quantities shown in Table 1. These features are calculated using a window of the time series so it called "time-domain features". We should highlight that this window includes the lane change event if it is corresponding to a lane change example. The second features set is the single side spectrum of a window of the time series around the lane change event.

The goal of the first set of experiments is to find a good window size ($2w+1$) for each of the features set. In order to do so, we used RF with 100 trees. We changed $w$ from 2 to 20. This corresponds to a window size from 0.4 sec to 4 sec. We used leave-one-out-cross-validation to divide the data into a training set and a testing set. Where in each fold we keep all the examples belonging to one trip as a test dataset and all other trips are used as the training dataset. The training dataset is used to build the RF model which is tested using the unseen testing dataset. Then the classification outputs for the examples in the testing trip are saved. For each trip, we repeat this sequence of training-testing sequence and save the classification outputs. Finally, the classification outputs of all trips (i.e. the whole dataset) and the ground truth labels are used to calculate the True Positive Rate (TPR) and the False Positive Rate (FPR) where lane change is coded as positive. As shown in the top panel of Fig 5, the best RF using the time domain features is at = 16, $w = 14$ or $w = 18$ which is corresponding to 3.2 sec, 3.4 sec, and 3.6 sec respectively.

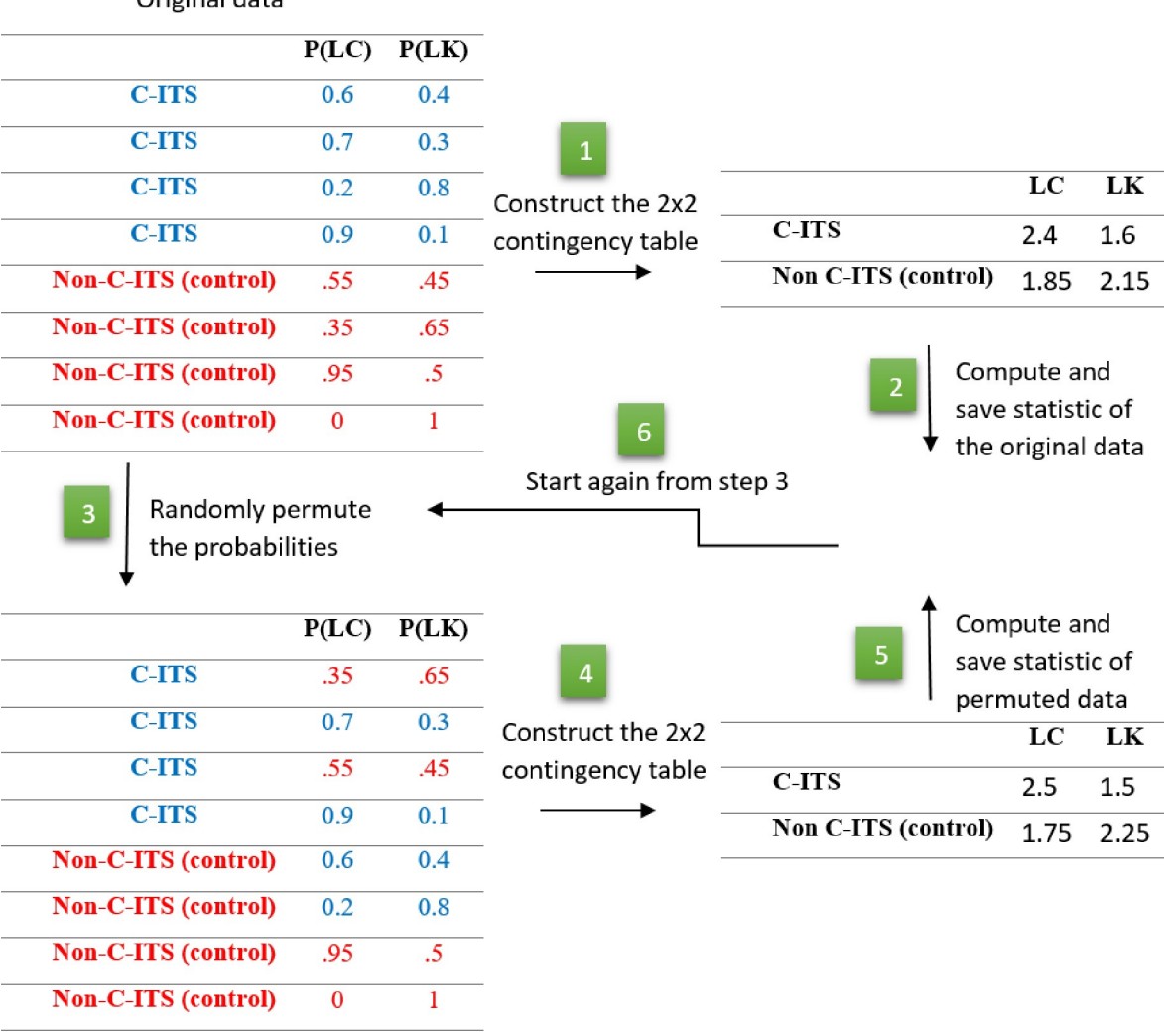

**Fig 4. Illustration of the permutation test.**

The bottom panel in Fig 5 shows the best RF using the frequency domain features is at $w = 16$, $w = 14$, $w = 18$, or $w = 20$ which is corresponding to 3.2 sec, 3.4 sec, 3.6 sec, and 4 sec respectively. It seems the frequency domain features are better than the time domain in terms of sensitivity. In addition, we decided to set $w = 15$ which equals 3 seconds window. We choose 3 seconds window because it has a relatively good TPR and FPR. In addition, because we are interested in the early lane change a 3 seconds window after the onset of the C-ITS warning is a reasonable choice, especially if we consider that the 85th percentile of the perception reaction time (PRT) of the drivers is in the range of 1.1 to 1.3 seconds [18, 19].

## Statistical testing

In this subsection, we show how the power of the adapted test change with the sample and effect size. We defined the effect size as $\pi_{\text{Non-C-ITS}} - \pi_{\text{C-ITS}} = \frac{M}{2M} - \frac{M+k}{2M} = \frac{-k}{2M}$ and the sample size is set equal to $4M$. The dataset has 3,994 lane changes examples and almost 8,000 lane-keeping examples. In addition to the ground truth labels of these examples, we have the hard

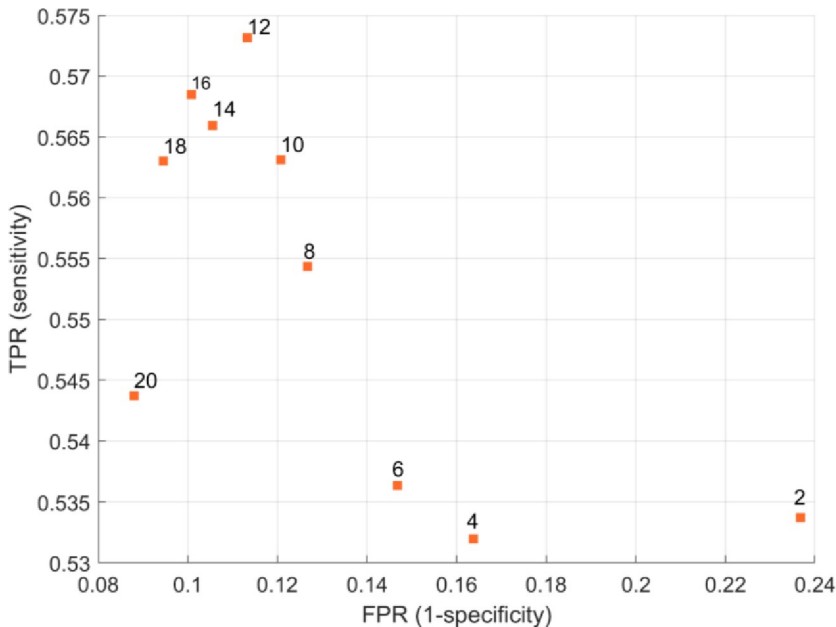

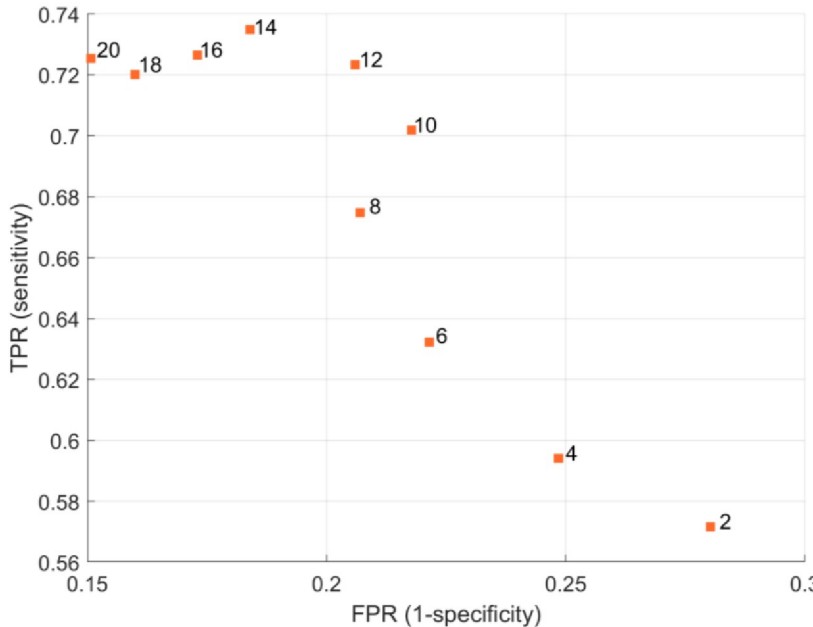

**Fig 5. The Receiver Operator Curve (ROC) of the RF using different window size where the top panel is the ROC of the RF using the time domain features and the bottom panel is the ROC of the RF frequency domain features.**

decision and soft decision classification output from the RF. We set up the experiment as shown in Table 3. We assume the sample size of the Non-C-ITS is $2M$ where randomly draw $M$ lane change examples and $M$ lane keeping examples without replacement. In the case of the C-ITS row, we randomly draw $M+k$ lane change examples and $M-k$ lane keeping examples without replacement. This makes the C-ITS examples equal the Non-C-ITS examples equal $2M$. Notice that, for the randomly chosen examples, if we summed the ground-truth label, it

**Table 3. The ground truth 2x2 contingency table used in this subsection.**

| | | Response | | |
|---|---|---|---|---|
| | | Lane change | Lane keeping | Total |
| Explanatory variable | C-ITS | $M+k$ | $M-k$ | $2M$ |
| | Non-C-ITS (control) | $M$ | $M$ | $2M$ |
| | | $2M+k$ | $2M-k$ | $4M$ |

would give us the numbers shown in Table 3. However, if we summed the hard decision or soft decision outputs of the RF of these examples, they would have different numbers than those shown in Table 3 due to the inaccuracy of the classifier.

The randomly selected subset of examples are used to estimate the p-value of the chi-square test as described in the following steps:

1. The constructed 2x2 contingency table from soft decision output of the RF is used to calculate the original chi-square statistic ($Stat^{org}$);

2. Randomly permute the subset of the examples as described in the Methods section;

3. Construct the 2x2 contingency table using the permuted data;

4. Calculate the and save the chi-square statistic of the permuted data ($Stat^{perm}$);

5. Repeat steps 2,3 and 5 $B$ times, where $B = 1000$;

6. Calculate the $p-value = \frac{\sum_{i=1}^{1000} I_{Stat^{perm} \geq Stat^{org}}}{1000}$ where $I$ is the indicator function.

We repeat the sequence of examples of the random draw, 2x2 contingency table construction, permutation test, and p-value calculation 1,000 times. Then we count the number of times we detected significant differences between C-ITS and Non-C-ITS given that there is a difference between the two groups. Besides, we count the number of times, we fail to detect significant differences between C-ITS and Non-C-ITS, given that there is no difference between the two groups. These counts are divided by 1,000 to estimate the $pr(reject\ H_0|H_1\ is\ true)$ and $pr(accept\ H_0|H_0\ is\ true)$ respectively.

The following figures show the $pr(reject\ H_0|H_1\ is\ true)$ and $pr(accept H_0|H_0\ is\ true)$ at different $M \in \{250,500,1000,1500\}$ and different effect size $\in [-0.2,0.2]$. In addition, we tried positive and negative k value in order to show that the ground truth odds ratio and the predicted odds ratio are in the same direction, as shown in Fig 6.

In addition to the $pr(reject\ H_0|H_1\ is\ true)$ and $pr(accept\ H_0|H_0\ is\ true)$ shown in the above figures, the probability of correct odds ratio are shown as well. The above curve shows only the probability of the ground truth odds ratio and the predicted odds ratio are both greater than one, or less than one without any statistical testing.

We conducted the following steps to test for the odds ratio:

1. The constructed 2x2 contingency table from soft decision output of the RF is used to calculate the original statistics $\theta^{org} = \frac{n_{11}n_{22}}{n_{12}n_{21}}$;

2. Randomly permute the subset of the examples as described in the methods section.

3. Construct the 2x2 contingency table using the permuted data;

4. Calculate the test statistics of the 2x2 contingency table of permuted data $\theta^{perm} = \frac{n_{11}^{perm} n_{22}^{perm}}{n_{12}^{perm} n_{21}^{perm}}$ and save it.;

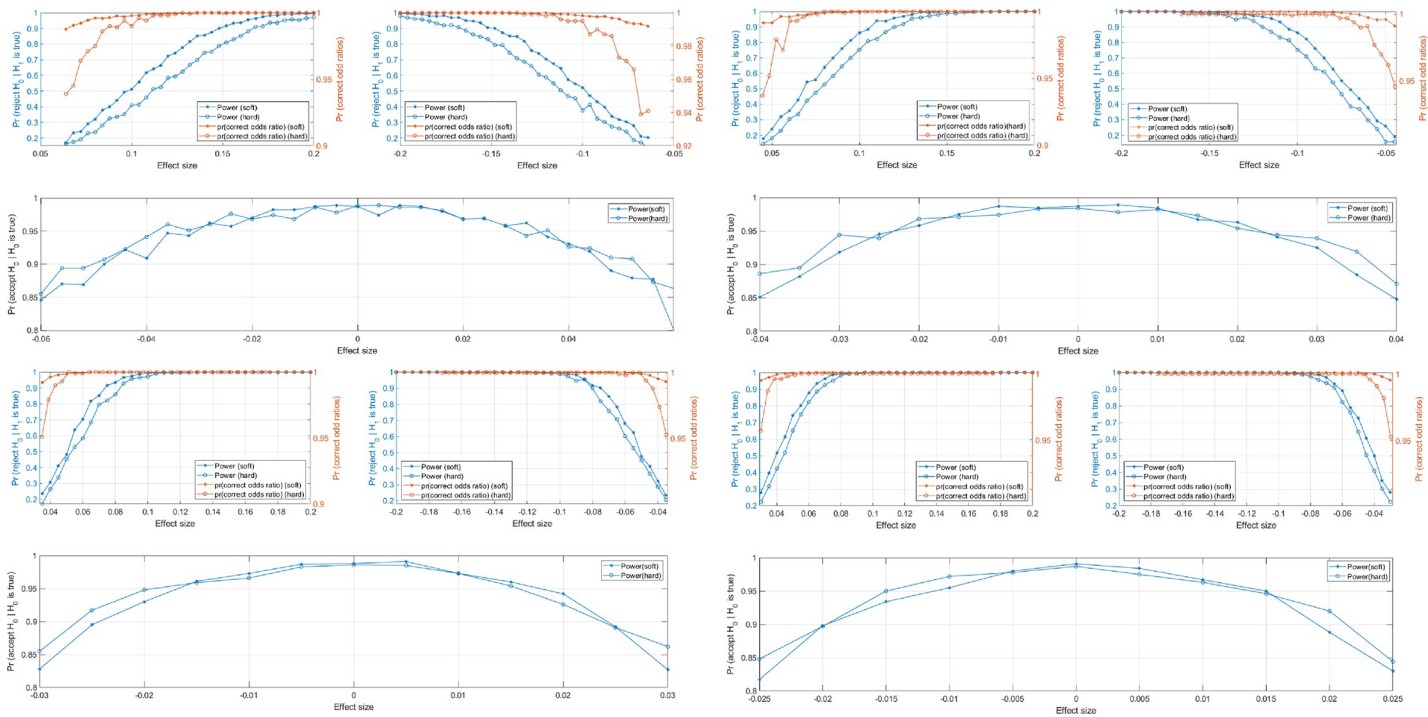

**Fig 6.** The $pr(reject\ H_0|H_1\ is\ true)$ and $pr(accept\ H_0|H_0\ is\ true)$ at (a) $M = 250$, (b) $M = 500$, $M = 1000$ and $M = 1500$.

5. Repeat steps 2,3 and 5 $B$ times, where $B = 1000$;

6. Calculate $p - value = \begin{cases} \dfrac{\sum_{i=1}^{1000} I_{\theta^{perm} \geq \theta^{org}} + \sum_{i=1}^{1000} I_{\theta^{perm} \leq \frac{1}{\theta^{org}}}}{1000} & \theta^{org} > 1 \\ \\ \dfrac{\sum_{i=1}^{1000} I_{\theta^{perm} \leq \theta^{org}} + \sum_{i=1}^{1000} I_{\theta^{perm} \geq \frac{1}{\theta^{org}}}}{1000} & \theta^{org} < 1 \end{cases}$ where $I$ is the indicator function.

We repeat the sequence of examples of the random draw, 2x2 contingency table construction, permutation test, and p-value calculation 1,000 times. Then we count the number of times the odds ratio is significantly different from one (i.e. C-ITS and lane change are dependent). Additionally we count the number of times we fail to reject the null hypothesis given that there is no difference between the two groups. These counts are divided by 1,000 to estimate the $pr(Correct\ association\ direction|H_1\ is\ true)$ and $pr(Correct\ association\ direction|H_0\ is\ true)$ respectively, as shown in Table 4

As shown in Table 4 above, pr(Correct association direction|$H_1$ is true) is high at the smallest sample size and reach one as the sample size gets bigger. In addition, the pr(Correct association direction|$H_0$ is true) is very high for all tested sample sizes at 0.02 effect size.

## Conclusion

In this paper, we proposed a framework to test the dependency of lane change and the C-ITS warning in two use-cases, namely: the roadworks warning (RWW) and slow/stopped vehicle (SSV). We used the longitudinal acceleration, lateral acceleration, roll, and yaw reading from the IMU. We extracted features from a short time window of predefined width. The extracted features from the four IMU time series were used to train an RF classifier to classify new

**Table 4. The $pr(Correct\ association\ direction|H_1\ is\ true)$ in the green cells and $pr(Correct\ association\ direction|H_0\ is\ true)$ in the blue cells.**

| Effect size | M = 250 | | M = 500 | | M = 1000 | | M = 1500 | |
|---|---|---|---|---|---|---|---|---|
| | Hard | Soft | Hard | Soft | Hard | Soft | Hard | Soft |
| 0.2000 | 0.9720 | 0.9970 | 1.0000 | 1.0000 | 1.0000 | 1.0000 | 1.0000 | 1.0000 |
| 0.1800 | 0.9370 | 0.9820 | 1.0000 | 1.0000 | 1.0000 | 1.0000 | 1.0000 | 1.0000 |
| 0.1600 | 0.8490 | 0.9400 | 0.9940 | 1.0000 | 1.0000 | 1.0000 | 1.0000 | 1.0000 |
| 0.1400 | 0.7440 | 0.8440 | 0.9720 | 0.9960 | 1.0000 | 1.0000 | 1.0000 | 1.0000 |
| 0.1200 | 0.5940 | 0.6790 | 0.9040 | 0.9700 | 1.0000 | 1.0000 | 1.0000 | 1.0000 |
| 0.1000 | 0.4160 | 0.5070 | 0.7400 | 0.8380 | 0.9720 | 0.9960 | 0.9980 | 1.0000 |
| 0.0800 | 0.2520 | 0.3140 | 0.5060 | 0.6320 | 0.8670 | 0.9480 | 0.9660 | 0.9970 |
| 0.0600 | 0.8580 | 0.8470 | 0.2860 | 0.3330 | 0.5850 | 0.7350 | 0.7920 | 0.8940 |
| 0.0400 | 0.9360 | 0.9420 | 0.8790 | 0.8570 | 0.2840 | 0.3190 | 0.3920 | 0.4940 |
| 0.0200 | 0.9740 | 0.9700 | 0.9570 | 0.9650 | 0.9350 | 0.9200 | 0.9100 | 0.9070 |
| 0 | 0.9900 | 0.9860 | 0.9850 | 0.9860 | 0.9870 | 0.9910 | 0.9840 | 0.9880 |
| -0.0200 | 0.9720 | 0.9760 | 0.9670 | 0.9610 | 0.9330 | 0.9240 | 0.9000 | 0.8970 |
| -0.0400 | 0.9370 | 0.9320 | 0.8740 | 0.8650 | 0.2530 | 0.3130 | 0.3970 | 0.4940 |
| -0.0600 | 0.8640 | 0.8390 | 0.2830 | 0.3700 | 0.6070 | 0.7100 | 0.7760 | 0.8850 |
| -0.0800 | 0.2550 | 0.3000 | 0.5250 | 0.6170 | 0.8870 | 0.9370 | 0.9750 | 0.9920 |
| -0.1000 | 0.3620 | 0.5110 | 0.7480 | 0.8570 | 0.9760 | 0.9980 | 0.9970 | 1.0000 |
| -0.1200 | 0.5860 | 0.7440 | 0.8960 | 0.9670 | 0.9960 | 1.0000 | 1.0000 | 1.0000 |
| -0.1400 | 0.7440 | 0.8410 | 0.9710 | 0.9890 | 0.9990 | 1.0000 | 1.0000 | 1.0000 |
| -0.1600 | 0.8660 | 0.9480 | 0.9940 | 0.9990 | 1.0000 | 1.0000 | 1.0000 | 1.0000 |
| -0.1800 | 0.9210 | 0.9730 | 0.9990 | 0.9990 | 1.0000 | 1.0000 | 1.0000 | 1.0000 |
| -0.2000 | 0.9720 | 0.9900 | 1.0000 | 1.0000 | 1.0000 | 1.0000 | 1.0000 | 1.0000 |

unseen feature vectors. In order to evaluate the dependency of the lane change and the C-ITS, we collected data right after the onset of the C-ITS warnings that are related to the above use cases. For each warning data, a window of a three-second of each time series was used to extract the features, which were concatenated to form one input example to the RF. The Non-C-ITS examples were collected the same way but with a turned-off HMI. Finally, the classification probabilities of all C-ITS and Non-C-ITS examples were used to build the 2x2 contingency table and adopted statistical tests were applied. The experimental results showed that using the soft decision as an input to construct the contingency table for the Chi-square test gives better statistical power and a higher probability of correct acceptance of the null hypothesis. In addition, using the soft decision as an input to construct the contingency table for the odds ratio test gives a better probability of correct association direction. Consequently, the proposed framework is suitable to check the lane change and C-ITS dependency.

## Supporting information

**S1 Dataset.**
(DOCX)

**S2 Dataset.**
(XLSX)

## Acknowledgments

The authors would like to acknowledge the support of the iMOVE Cooperative Research Centre (CRC) for this work.

## Author Contributions

**Conceptualization:** Mohammed Elhenawy, Sébastien Glaser, Andy Bond, Andry Rakotonirainy, Sébastien Demmel, Mahmoud Masoud.

**Data curation:** Sébastien Demmel.

**Funding acquisition:** Andry Rakotonirainy.

**Investigation:** Mohammed Elhenawy, Andy Bond, Andry Rakotonirainy, Mahmoud Masoud.

**Methodology:** Mohammed Elhenawy, Sébastien Glaser.

**Software:** Mohammed Elhenawy.

**Supervision:** Andy Bond, Andry Rakotonirainy.

**Validation:** Mohammed Elhenawy, Mahmoud Masoud.

**Visualization:** Mohammed Elhenawy.

**Writing – original draft:** Mohammed Elhenawy, Mahmoud Masoud.

**Writing – review & editing:** Sébastien Demmel, Mahmoud Masoud.

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
