## [Decision Letter · Decision Letter 0]

4 Dec 2019

PONE-D-19-28884

A Framework for Testing Independence between Lane Change and Cooperative Intelligent Transportation System

PLOS ONE

Dear Dr. Mahmoud Masoud,

Thank you for submitting your manuscript to PLOS ONE. After careful consideration, we feel that it has merit but does not fully meet PLOS ONE’s publication criteria as it currently stands. Therefore, we invite you to submit a revised version of the manuscript that addresses the points raised during the review process.

We would appreciate receiving your revised manuscript by Jan 18 2020 11:59PM. To enhance the reproducibility of your results, we recommend that if applicable you deposit your laboratory protocols in protocols.io, where a protocol can be assigned its own identifier (DOI) such that it can be cited independently in the future. For instructions see: http://journals.plos.org/plosone/s/submission-guidelines#loc-laboratory-protocols

We look forward to receiving your revised manuscript.

Kind regards,

Chen Lv, PhD

Academic Editor

PLOS ONE

Journal Requirements:

3. Please ensure that you refer to Figures 1-3, 5 and 9 in your text as, if accepted, production will need this reference to link the reader to the figure.

Additional Editor Comments:

Please carefully address every comment from reviewers.

Reviewers' comments:

Reviewer's Responses to Questions

**Comments to the Author**

1. Is the manuscript technically sound, and do the data support the conclusions?

Reviewer #1: Yes

Reviewer #2: Yes

2. Has the statistical analysis been performed appropriately and rigorously? 

Reviewer #1: Yes

Reviewer #2: Yes

3. Have the authors made all data underlying the findings in their manuscript fully available?

Reviewer #1: Yes

Reviewer #2: Yes

4. Is the manuscript presented in an intelligible fashion and written in standard English?

Reviewer #1: No

Reviewer #2: Yes

5. Review Comments to the Author

Reviewer #1: This work proposed a framework to test the dependency between lane change and C-ITS with trained random forest as feature extractor.

Some concerns:

1. Line 18: however; ?

2. L 30: too many specific countries.

3. L 37: what do CEN, ISO, and ETST stand for?

4. L 46: Is it necessary to list many projects?

5. L 72: what does LIDAR stands for?

6. Please keep the reference style consistent through the whole script.

7. L 109; state of THE art.

8. L 112: proposed

9. L123: .,

10. L128: use abbreviated version.

11. So many existing works are listed with excess details

12. The weakness of existing works

13. Why the proposed framework is better to others.

14. L138: a classifier for what? Should be specific.

15. L 140: hard to understand.

16. L 143: statement is too general.

17. Use abbreviated versions after explained. For example L 164 and many other places.

18. L224: Description of dataset is too long.

19. L 252: a noun is not a sentence.

20. How could random forest DOES NOT suffer from overfitting?

21. L 274: Have the authors proofread this script before submitting?

22. L 277: of and?

23. L 284: its length?

24. L 345: dds?

25. L 510: use words rather than variables in conlusion.

Reviewer #2: The revised manuscript "A Framework for Testing Independence between Lane Change and Cooperative Intelligent Transportation System" has addressed my earlier comments. I have no other comments to submit.

6. PLOS authors have the option to publish the peer review history of their article (what does this mean?). If published, this will include your full peer review and any attached files.

Reviewer #1: No

Reviewer #2: No

---

## [Author Response · Author response to Decision Letter 0]

7 Jan 2020

Response Letter for “PONE-D-19-28884”

Reviewer#1, Concern # 1: Line 18: however?

Author response: Thank you, fixed.

Reviewer#1, Concern # 2: L 30: too many specific countries.

Author response: Thank you. We reduced the number of the specific countries that we mentioned as an example of countries involved in the research and development of ITS.

Reviewer#1, Concern # 3: L 37: what do CEN, ISO, and ETST stand for?

Author response: Thank you, fixed. We replaced “The interaction between ITS components is based on several standards such as CEN[2], ISO[3], ETST [4], depending on the jurisdiction.” By “The interaction between ITS components is based on several standards developed by different organizations such as European Committee for Standardization (CEN) [2], the International Organization for Standardization (ISO) [3], the European Telecommunications Standards Institute (ETST) [4], depending on the jurisdiction.”

Reviewer#1, Concern # 4: L 46: Is it necessary to list many projects?

Author response: Thank you. To our mind, listing these projects is important to show the widespread of these projects around the world. Moreover, the interested reader can find references to these important projects.

Reviewer#1, Concern # 5: L 72: what does LIDAR stand for?

Author response: Thank you. LIDAR stands for “LIght Detection and Ranging”. We updated the paper accordingly 

Reviewer#1, Concern # 6: Please keep the reference style consistent through the whole script.

Author response: Mahmoud task PLEASE CHECK

Reviewer#1, Concern # 7: L 109; state of THE art.

Author response: Thank you, fixed 

Reviewer#1, Concern # 8: L 112: proposed

Author response: Thank you, fixed 

Reviewer#1, Concern # 9: L121: remove,.

Author response: thank you, fixed. 

Reviewer#1, Concern # 10: L128: use the abbreviated version.

Author response: Thank you, fixed. We replaced “the three raw physiological signals” by “the ECG, GSR and RR signals.”

Reviewer#1, Concern # 11: So many existing works are listed with excess details

Author response: We mentioned much-existing work with some details to show that the previous work used data modalities that are not available in our FOT.

Reviewer#1, Concern # 12: The weakness of existing works

Author response: When we described much existing work, our goal was to clarify that they used different data modalities that are not available in our case. Moreover, to the best of our knowledge, we are the first who proposed using data collected from the C-ITS and the IMU to test the independence between lane change and cooperative intelligent transportation system without using other data modalities such as videos.

Reviewer#1, Concern # 13: Why the proposed framework is better for others.

Author response: As mentioned at the end of the introduction section “The present paper is different from earlier research in that it developed a framework consisting of the following:

1- Training a lane change/lane-keeping machine learning classifier using a limited number of sensing modalities (i.e. longitudinal acceleration, lateral acceleration, roll, and yaw)

2- Overcoming the uncertainty of machine learning classifier if it classifies one data point by summing up the output probabilities of the trained classifier of the whole collected data (i.e. all data points) to construct the 2x2 contingency table. 

3- Estimating the null hypothesis of the test statistic using the proposed permutation test.

This framework enables us to test the independence between the lane change and the C-ITS warning without collecting expensive data modalities such as videos to train accurate machine learning classifier. In other words, the proposed framework allows us to test the hypotheses that assume C-ITS warnings at safety-critical situation help the drivers changing lane safely compared to the drivers who do not receive C-ITS warnings.”

Reviewer#1, Concern # 14: L138: a classifier for what? Should be specific.

Author response: Thank you, fixed. A classifier for lane change/lane-keeping. 

Reviewer#1, Concern # 15: L 140: hard to understand.

Author response: We replaced the word pooling by the word summing up to make the statement more clear. 

Reviewer#1, Concern # 16: L 143: statement is too general.

Author response: We made it more specific “This framework enables us to test the independence between the lane change and the C-ITS warning without collecting expensive data modalities such as videos to train accurate machine learning classifier. In other words, the proposed framework allows us to test the hypotheses that assume C-ITS warnings at safety-critical situation help the drivers changing lane safely compared to the drivers who do not receive C-ITS warnings.”

Reviewer#1, Concern # 17: Use abbreviated versions after explained. For example, L 164 and many other places.

Author response: Thank you, fixed. 

Reviewer#1, Concern # 18: L224: Description of dataset is too long.

Author response: Thank you, fixed. We reduced the description of the dataset 

Reviewer#1, Concern # 19: L 252: a noun is not a sentence.

Author response:

Reviewer#1, Concern #20: How could a random forest DOES NOT suffer from overfitting?

Author response: Thank you. In the random forest subsection, we stated that “The most significant advantage of the RF approach is that it does not suffer from overfitting because of the Law of Large Numbers, where the error rate of RF will get smaller as more CARTs are added. By adding more CATRs, the error rate will be saturated but will never get larger no matter how many CARTs are added.”

Reviewer#1, Concern #21: L 274: Have the authors proofread this script before submitting?

Author response: Yes, thank you.

Reviewer#1, Concern #2222. L 277: of and?

Author response: thank you, fixed

Reviewer#1, Concern #23: L 284: its length?

Author response: thank you,fixed

Reviewer#1, Concern #24: L 345: dds?

Author response: thank you, fixed

Reviewer#1, Concern #25: L 510: use words rather than variables in conclusion.

Author response: thank you, fixed.

Reviewer #2: The revised manuscript "A Framework for Testing Independence between Lane Change and Cooperative Intelligent Transportation System" has addressed my earlier comments. I have no other comments to submit.

 Author response: thank you. ________________________________________

 6. PLOS authors have the option to publish the peer review history of their article (what does this mean?). If published, this will include your full peer review and any attached files.

If you choose “no”, your identity will remain anonymous, but your review may still be made public.

Do you want your identity to be public for this peer review? For information about this choice, including consent withdrawal, please see our Privacy Policy.

Reviewer #1: No

Reviewer #2: No

---

## [Decision Letter · Decision Letter 1]

4 Feb 2020

A Framework for Testing Independence between Lane Change and Cooperative Intelligent Transportation System

PONE-D-19-28884R1

Dear Dr. Mahmoud Masoud,

We are pleased to inform you that your manuscript has been judged scientifically suitable for publication and will be formally accepted for publication once it complies with all outstanding technical requirements.

With kind regards,

Chen Lv, PhD

Academic Editor

PLOS ONE

Additional Editor Comments (optional):

Thanks for your revision.

Reviewers' comments:

Reviewer's Responses to Questions

**Comments to the Author**

1. If the authors have adequately addressed your comments raised in a previous round of review and you feel that this manuscript is now acceptable for publication, you may indicate that here to bypass the “Comments to the Author” section, enter your conflict of interest statement in the “Confidential to Editor” section, and submit your "Accept" recommendation.

Reviewer #1: (No Response)

Reviewer #2: All comments have been addressed

2. Is the manuscript technically sound, and do the data support the conclusions?

Reviewer #1: Yes

Reviewer #2: Yes

3. Has the statistical analysis been performed appropriately and rigorously? 

Reviewer #1: Yes

Reviewer #2: Yes

4. Have the authors made all data underlying the findings in their manuscript fully available?

Reviewer #1: Yes

Reviewer #2: Yes

5. Is the manuscript presented in an intelligible fashion and written in standard English?

Reviewer #1: Yes

Reviewer #2: Yes

6. Review Comments to the Author

Reviewer #1: This resubmitted A Framework for Testing Independence between Lane Change and Cooperative Intelligent Transportation System addressed almost all my previours concerns. I have no other comments.

Reviewer #2: (No Response)

7. PLOS authors have the option to publish the peer review history of their article (what does this mean?). If published, this will include your full peer review and any attached files.

Reviewer #1: No

Reviewer #2: No

---

## [Editor Report · Acceptance letter]

7 Feb 2020

PONE-D-19-28884R1 

A Framework for Testing Independence between Lane Change and Cooperative Intelligent Transportation System 

Dear Dr. Masoud:

I am pleased to inform you that your manuscript has been deemed suitable for publication in PLOS ONE. Congratulations! Your manuscript is now with our production department. 

With kind regards,

on behalf of

Dr. Chen Lv 

Academic Editor

PLOS ONE